# Malaria prevention in the age of climate change: A community survey in rural Senegal

Andrew C. L. Sherman[ID][1,2*], C. Andrew Aligne[1], Jesse D. Matthews[3,4]

**1** University of Rochester Medical Center Department of Pediatrics and Hoekelman Center, Rochester, New York, United States of America, **2** Netlife, Fairport, New York, United States of America, **3** St. Charles Medical Center, Bend, Oregon, United States of America, **4** Swedish Heath Services, Seattle, Washington, United States of America

\* shermaac@gmail.com

## Abstract

### Background

Malaria results in over 600,000 deaths per year, with 95 percent of all cases occurring in sub-Saharan Africa. Insecticide treated mosquito nets have long been proven to be the most effective prevention method to protect at-risk people from malaria. Temperature increases may now be changing sleeping habits and how people use available mosquito nets. Based on observations of increasing outdoor sleeping and fragility of the mosquito nets, this study evaluated a rural west African population to determine barriers to mosquito net use, including net fragility, heat and outdoor sleeping.

### Methods

This study used a social ecological framework used by the Peace Corps to determine this community's barriers to malaria prevention. We practiced community-based participatory research by developing and implementing a survey in rural southeast Senegal. Local village health workers received special training to implement this survey. Observations of the mosquito nets and sleeping spaces were performed by surveyors. 164 households in 20 villages were surveyed from October to November of 2012.

### Results

There was a 100% response rate, with 164 of the 164 selected households surveyed, representing 21% of this local population. For the 1806 family members, respondents assessed a total need of 1565 nets, implying that each individual in this area needs 0.86 nets (95% CI: 0.77–0.95). Survey responses gave rich, informative responses about mosquito net use. For example: 'If it's in the room set up properly under the mattress then it will be fine. But if it's outside with the beds that don't have

**Data availability statement:** All surveys and raw data files are available from the University of Rochester Research Repository database (DOI https://doi.org/10.60593/ur.d.28533629).

**Funding:** The author(s) received no specific funding for this work.

**Competing interests:** The authors have declared that no competing interests exist.

mattresses, then it will deteriorate quickly.' The main reasons for not using an available net were heat and fragility of the nets. This population had very positive attitudes regarding mosquito nets and appreciated the work of local malaria educators.

## Conclusions

In a rural Senegalese population with a high malaria burden, our survey indicated a need for 0.86 insecticide treated nets per person. This is 54% higher than the current WHO recommendation of 0.56 ITNs per person. Our findings suggest that there are not enough nets because routine village conditions lead to considerable net damage, and because the heat leads people to sleep outdoors, where they likely do not have mosquito nets. With global warming, we suspect this population will spend even more time sleeping outside, aggravating the current insufficiency. Further research should investigate optimal interventions to address this challenge, including nets designed for outside use and for higher durability.

## Introduction

Malaria results in over 600,000 deaths per year globally. Approximately 95 percent of all malaria cases and deaths occur in sub-Saharan Africa, with 80% of those deaths in children under 5 years of age. Significant reductions in malaria mortality have occurred since 2000 [1]. These successful campaigns were primarily focused on the mass distribution of insecticide treated mosquito bednets (ITNs). These efforts also added anti-malaria methods like indoor residual insecticide spraying (IRS), rapid diagnostic tests and improved medical treatments. ITN access remains the priority of malaria prevention efforts as they are inexpensive, long-lasting and relatively easy to use. These nets are traditionally attached on indoor structures to hang over beds. They prevent mosquitoes, which transmit the malaria parasite, from biting vulnerable family members while they sleep. ITNs have long been proven to the be most effective prevention method to protect people from malaria [2–4]. Providing ITNs to communities in malaria endemic regions reduces all-cause child mortality by 17 percent [5].

Despite these interventions, malaria continues to be a significant problem around the world. In countries like Senegal, these measures have helped, but malaria is still a main cause of death in children [6–7]. This study took place in the region of Kedougou, in rural southeastern Senegal, where there is high malaria prevalence despite wide availability of ITNs [8].

Based on our experiences distributing thousands of nets in this region, we believed that the people were highly educated about malaria and very receptive to ITNs. Anecdotal observations from living in one village there led us to suspect that the available nets were not providing optimal protection because of factors that made them impractical for long-term daily use. In particular, we noted that sharp edges on bedding materials damaged the nets, shortening their useful lifespan. In addition, we saw many people sleeping outdoors, where they had no nets, to escape the stifling

heat inside their huts. If outdoor sleeping because of heat is a significant factor impeding the prevention of malaria, then climate change will be exacerbating this problem. Global warming is already affecting health around the world, with vulnerable populations disproportionately suffering from its impacts [9]. Sub-Saharan Africa is not spared from this phenomenon. Over the last century, the mean temperature in southeast Senegal has steadily increased, with the average daily temperature increasing from 27.5 Celsius (81.5 Fahrenheit) in 1929 to 29.7 (85.5) in 2023 [10]. This warming favors conditions for mosquito population growth and malaria transmission.

We thus thought it would be important to determine whether our observations in one village were generalizable to the region. The goal of the study was to identify social and environmental factors that could be impeding malaria control by diminishing effective net usage. In addition to outdoor sleeping and net damage, we considered the knowledge and perceptions of malaria and ITNs, and community health priorities. The intended outcome of this research is to provide actionable information to adapt and maximize access to evidence-based malaria prevention.

The primary hypothesis of this study was that for adequate protection from malaria-bearing mosquitoes, people in this region need more than the officially recommended 0.56 nets per person. We further hypothesized that net fragility and outdoor sleeping were major factors contributing to the insufficiency of nets.

## Materials and methods

### Study population

This study was conducted in the rural region of Kedougou (12.32° N, 12.18° W), located in the southeast corner of Senegal. The population of Senegal is 18 million, and the population of Kedougou is 245,000 [11–12]. The rainy season extends from May to November [13]. Kedougou has a poverty rate of 71.3% and an adult illiteracy rate of 58% [12]. Kedougou is composed of four sub-regions referred to as arrondissements. Our research focused on the arrondissement of Bandafassi and the 37 villages served by its health post. The village populations ranged from 30 to 952 people. This is a rural population, and most families have a livelihood of agriculture and cattle herding [11]. Over 95% of the population is Muslim, with the remaining practicing Christianity or animism [14]. The main ethnic groups in this sub-region are Peul, Mandinka, Bedik, and Bassari. French is taught throughout the school systems, so many residents communicate well in French. The average household size is larger than other areas in West Africa [15]. The standard household has 8 people and larger "collective" families average more than 15 people [11–12]. This area was selected because it has been a zone of high malaria morbidity and mortality.

### Study framework

This study's research question asked what were the barriers to using an available ITN for people in rural Senegal? A secondary question of this research is how can we quantify the identified barriers and report them in an actionable way that further improves malaria prevention? To address these questions, this study used a framework (Fig 1) used by the Peace Corps specifically to look at malaria control in rural sub-Saharan Africa [16]. This framework combines important aspects of previous models and incorporates a state of sustainability [17]. The Peace Corps framework has potential to guide effective interventions, based on a community's assessment of barriers to regular ITN use.

We practiced community-based participatory research by incorporating local stakeholders into the research team. Input from these community members informed the study's design, implementation, analysis and interpretation. Our survey questions were adapted from the 2008–2009 Senegal Malaria Indicator Survey [8]. A range of open-ended questions were added to the survey to assess key components of behavior, with special attention to barriers of ITN use. We also encouraged the surveyors to include observations and relevant comments on the survey. Survey questions are available for review in the appendix (S1-S3 Files).

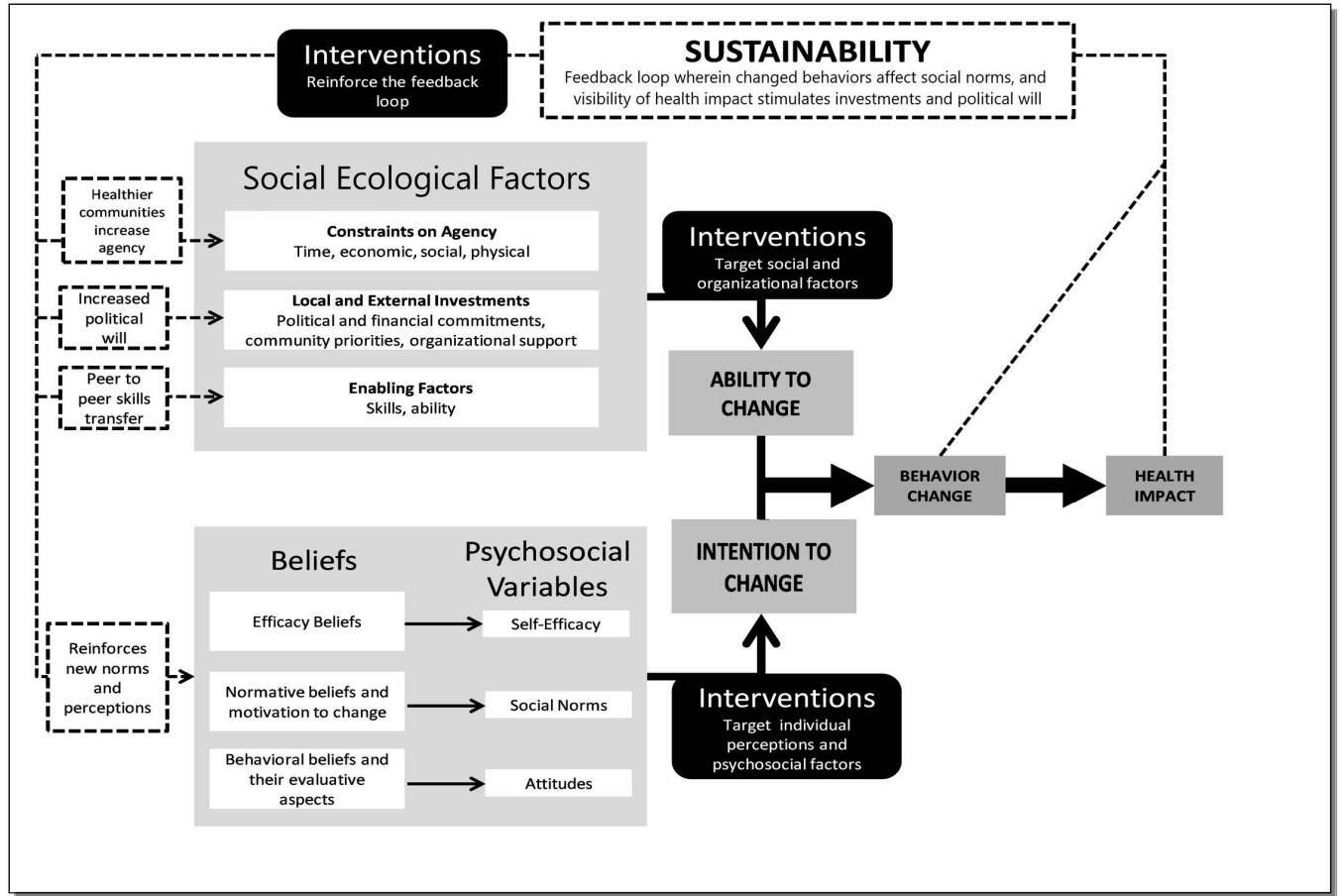

**Fig 1. Study theoretical model.** Adapted from McLaughlin and Peace Corps. This model integrates important social ecological factors and applies them to rural Senegal malaria prevention.

## Sampling and data collection

The study acquired preliminary census data from the Bandafassi health post to determine village population size. To confirm current population numbers, we also obtained village census data from each of the village chiefs served by the Bandafassi health post. We performed a simple random sample to select households, using a random number generator (Random~Number, PocketLab). We selected between 4 and 20 households per village, or between 15–33% of all households, depending on the size and location of the village. In all, we selected 164 households from 20 villages. Each questionnaire was labeled with a random unique identifier number.

In Senegal, the research team met with the coordinating Bandafassi health post nurse, as well as the village chiefs from sampled villages. The village chiefs agreed to partner with the study team, providing up-to-date census data and verification that questionnaires were properly administered by the surveyors. The questionnaire was piloted in a village that was not part of the study sample. Based on feedback from piloting the survey, we adapted the questionnaire for local comprehension and cultural relevance. The study used local surveyors, recommended by the health post nurse, to administer the survey. Surveyors were selected based on their experience with similar surveys and their bilingualism in French and a local language. The study employed eleven surveyors: 7 men and 4 women. The research team and health post

nurse developed a training curriculum for the surveyors. After the training, the surveyors demonstrated enhanced abilities to implement the questionnaire in a culturally sensitive manner.

The surveyors conducted 164 structured interviews in October and November of 2012. Data collection coincided with the last months of the rainy season, when at-risk people could have been decreasing their net use. Surveyors were assigned to villages according to their language skills. Surveys were conducted in the homes of participants and directed at the heads of households, with input from available family members. Participants were not paid.

### Data analysis

Surveyors recorded responses for each questionnaire verbatim in the language used by the participant. If a response was in a local language, the multilingual interviewer would immediately translate and write the response again in French. Translations were made from French to English by a bilingual research team member. English translations were transcribed into a REDCap interface. For descriptive analysis of responses to open-ended questions, three researchers reviewed all responses and came to a consensus on which categorical codes best represented the response themes. The codes were also chosen based on their potential to reflect known categories of ITN-use barriers and the themes of the selected theoretical framework (Fig 1). Analysis was used to address the gaps in the present literature, specifically the knowledge, perceptions and cues to action of ITN use. Four independent researchers assigned codes for each survey response. The team discussed disagreements until a consensus interpretation was agreed upon.

A section designed to quantify family members and sleeping spaces from the head of the household and surveyor observations was also evaluated. The research team made field observations of mosquito nets and village beds. Surveyors were instructed to estimate that one ITN was needed for each sleeping space. Members took detailed notes on these observations, and then repeatedly met to discuss possible interpretations. Senegalese members of the research team verified interpretations within the context of the local culture. Incomplete responses were excluded from the analysis. Data is available for review through the University of Rochester Research Repository, DOI 10.60593/ur.d.28533629.

### Ethical considerations

The University of Rochester Research Subjects Review Board evaluated all aspects of this project, including the consent process, and determined that it met federal and university criteria for exemption (RSRB00041566). The Bandafassi health post administration approved and partnered with this project for the formulation of aims, objectives and the training of the surveyors. We met with all the village chiefs of the selected villages, and they approved of this project. Despite this study being exempt, low risk to participants, and having the data analyzed anonymously, we still had each participant give verbal consent before the questionnaire. Each verbal consent was performed in the local language by the surveyor. The script of the verbal consent was modified from the 2005 Malaria Indicator Survey, which was developed by the Monitoring and Evaluation Working Group of Roll Back Malaria [18]. The village chief of each participant served as witness that all aspects of each survey were properly performed, including the verbal consent. To document proper verbal consent, each surveyor signed the survey and each village chief imprinted the survey with the official governmental village stamp. No personal identifiers were recorded. Additional information regarding the ethical, cultural, and scientific considerations specific to inclusivity in global research is included in the Supporting Information (S1 Checklist).

### Results

All 164 selected households were successfully surveyed, giving a 100% response. Most individual questions had responses over 96%. The question with the lowest response had a 93% response. The 164 households represented 1806 people or about 21% of the total population of the Bandafassi region. 152 heads of household gave updated information on household size.

The 20 selected villages were an average of 9.7 kilometers away from the nearest staffed health care facility. The average distance from a health care facility for all 37 area villages was 9.3 kilometers. The 20 selected villages had an average population of 350 people. The average population per village of all 37 area villages was 235 people. The average household size was 11.9 people.

### Reported need for nets

97.6% of households had access to at least one net. For the 1806 family members, respondents assessed a total need of 1565 nets. This implies that each individual needs 0.86 nets (95% CI: 0.77–0.95), i.e., the 12,441 people in this region need a total of 10,678 nets.

### Barriers to optimal net utilization and malaria control

**Heat and outdoor sleeping.** When asked "When you think about mosquito nets, what do you dislike about them?" heat or lack of airflow was commonly discussed.

*'When the weather is hot, it keeps the air from getting through and makes it hard to sleep comfortably.'*

*'[T]he mosquito nets make it hot and suffocate us.'*

When asked "There are people that do not use a mosquito net every day. In your opinion, why do these people not use a net every day?" a respondent discussed the mosquito net's effect on airflow:

*'During the hot weather, some people say that the mosquito net makes it even hotter.'*

When asked "How could the mosquito net be improved?" another respondent discussed how outdoor sleeping can cause accelerated damage of nets:

*'If it's in the room set up properly under the mattress then it will be fine. But if it's outside with the beds that don't have mattresses, then it will deteriorate quickly.'*

When asked if they had anything to add, this respondent discussed a sense of vulnerability with being outside in the evening:

*'Mosquito nets are effective. Except that when we are outside before going to bed, the mosquitos that bite us give us malaria.'*

A surveyor observed situations where children were vulnerable during outdoor sleeping:

*'There were miradors* [elevated sleeping platforms] *outside but no mosquito nets. Some children were sleeping there. Others on the other hand were sleeping inside where all the beds had nets hanging.'*

**ITN fragility.** A majority of respondents described ITN fragility as a reason for needing more nets. When asked: "How could the mosquito net be improved?" 62.8% of respondents discussed a need for increased durability:

*'They need to be made tougher and longer-lasting.'*

*'We have cots made of pleated wooden slats; we would like the nets to be improved so that they are more appropriate for our beds.'*

Also, when asked: "Are all the mosquito nets you received at the last distribution still here?" 23.7% said no. When a follow up question asked: "What happened to the other nets?" 69.1% mentioned damage.

**Overwashing.** When asked what happened to the missing nets, respondents described concerns about how the washing process could damage the nets:

*'The mosquito nets lasted, but then got ruined. The women wash them and dry them under the hot sun and so they wind up getting ripped.'*

Some respondents viewed the chemicals as something that causes illness and thus needed to be cleaned or removed:

*'There are mosquito nets that have an odor that causes colds, so we wash them before using them.'*

## Facilitators of net usage and malaria control

**Acceptability of community malaria educators.** When discussing other community priorities, participants often valued the role of education. When asked "If there where health aides who were trained to help solve problems with mosquito nets and malaria in general would you be interested in having someone come to the household to help you?" 98.2% said yes. 91.3% thought this malaria educator should visit their homes two or more times a year. 52.8% thought they should visit 4 or more times a year, regularly suggesting even more frequent visits from "*three times during the rainy season*" to "*all the time*".

When asked "In your opinion, what do you think the people of the village could do to prevent more cases of malaria? You can talk about things other than mosquito nets", education was discussed regularly:

*'What the people of the village could do to help prevent malaria is to organize educational sessions and conversations all the time.'*

*'We should have conversations every month, if that's possible. Especially before, during, and after the rainy season.'*

**Acceptability of ITNs.** When asked "When you think about mosquito nets, what do you like about them?" 100% of respondents expressed positive attitudes toward nets, including protective benefits and comfort. They also gave examples of perceived risks of malaria:

*'[H]anging up mosquito nets is better than having to throw away all of one's money on prescriptions.'*

*'If we didn't have mosquito nets, we would have people falling ill all the time; that is why I always tell the children to lower the mosquito nets when they go to bed.'*

When asked "There are people that do not use a mosquito net every day. In your opinion, why do these people not use a net every day?" 63.1% mentioned the words "negligence or ignorance":

*'People who don't use a mosquito net every day in my opinion are people who don't know the risks they are taking.'*

*'People who do not use mosquito nets every day, in my opinion, it is because of laziness or negligence...'*

**Suggestions from the community.** When asked about possible community actions to reduce malaria, interventions about cleaning were by far the most common response (60.4%).

*'The people of the village could help with the prevention of malaria by cleaning up the village properly.'*

*'Avoid pools of stagnant water, destroying them during the rainy season. Everything that could contain water or otherwise bring mosquitoes should be destroyed.'*

## Discussion

In a rural Senegalese population with a high malaria burden, our survey found a need of 0.86 nets per person. This is 54% higher than the current WHO recommendation of 0.56 nets per person [19]. The WHO recommendation for the number of nets per person has existed since 2010. The WHO suggests the "ratio can be adjusted as needed if there are data that support such adjustment" [20]. It has been suggested that the ratio calculation may need to be adjusted for the early damage of nets and its effect on ITN retention. This study's findings suggests that the ratio should be adjusted further to account for the substantial degree of outdoor sleeping.

A major strength of this study is that the lead researcher lived in this area as a Peace Corps volunteer and established positive relationships with community leaders. These connections continued for many years via the public health work of the non-profit organization, Netlife. These past partnerships and the use of local languages helped create a strong, multifaceted study team. Pilot testing helped to create a culturally accepted final version of the survey. The trust from these past relationships and the culturally sensitive administration of the surveys contributed to the 100% response.

A potential limitation of our data is that the number of nets needed by families was based on the head of household's assessment. It would be more objective to observe all household members and how they use their sleeping spaces at night, however it is hard to imagine how to carry that out practically or ethically. Nevertheless, there could be a concern that heads of household would ask for more nets than necessary. To address this, we analyzed open-ended direct observations by the surveyors. Overall, when compared with these observations, the participant responses showed some underestimates, some overestimates and many appropriate estimates of requested nets. So, we found there was no systematic bias of heads of household overestimating their families' ITN needs. Also, the surveyor observations did not take into account potential outside sleeping spaces; doing so would have pushed all the participant responses in the direction of being underestimates of need for nets.

Our survey was conducted in 2012 and so could now be outdated. However, members of our team have visited these villages multiple times since then (as recently as 2023), and the situation with heat and outdoor sleeping has only become more pronounced. We therefore believe it now requires urgent attention, as one can expect that global warming will make the problem even worse. We have made regular observations of outdoor sleeping. Most household compounds have outdoor structures used during the day for shade or for food preparation. During the evenings, many family members drink tea and talk on these structures. Because of the heat, many people prefer to sleep the first part of the night outdoors, because there is better airflow outdoors than in the huts. When cool enough, most family members wake up and move to the sleeping spaces inside the huts. Regularly, family members would sleep one-third to one-half of the night outside. Although we observed mosquito nets being used on some outdoor structures (S1–S4 Figs), this was rare. Most household members only have enough nets for their indoor sleeping spaces.

The regularity with which these rural Senegalese people discussed heat/outdoor sleeping in their survey responses is consistent with this being an important issue. Because outdoor sleeping increases the number of sleeping spaces used by each family, it increases the number of nets needed to provide protection from mosquitos. In addition, if nets are used outside, they wear out faster. For example, the structures used for outdoor sleeping tend to have many sharp points, and nighttime winds cause the nets to regularly catch on these points. (S4 File) Very few ITNs are truly designed for outdoor spaces [21,22]. Consistent with Killeen and Grand Challenges in Global Health, this study supports the need to expand interventions to the outdoors [22,23].

Prior to this survey, few studies had specifically investigated outdoor sleeping as a barrier to ITN use. Studies conducted near Lake Victoria and in Burkina Faso found that outdoor sleeping was infrequent [24,25]. In contrast, we found

outdoor sleeping to be extremely prevalent, consistent with reports from Niger and Iran [26,27]. Heat has been found to be a significant barrier to ITN use, and causes this study population to regularly sleep outside [28–31].

ITN use is a cornerstone of malaria prevention, and if increasing barriers exist, it will be difficult to maintain past successes to fight malaria. Despite significant steady reductions in malaria mortality from 2000 to 2015, there has been a recent resurgence. The estimated 2025 Africa death rate is now 51.8 people per 100,000, whereas the previous prediction for 2025 had been 15.9 [1,32]. While there are many possible explanations for this underperformance, the increase in outdoor sleeping does not help the situation. We should provide more nets for people who sleep outside, and research should be conducted to develop modified nets more suitable for outdoor use.

If outdoor sleeping increases with rising temperatures, it could affect other evidence-based interventions like indoor residual spraying (IRS). In the past, it has been proven that the use of ITNs and IRS together is more protective against malaria than either intervention alone [33]. Additional research may be required to determine the effect of IRS to protect people sleeping on outdoor structures [34,35]. As in previous studies, this population expressed concerns about the durability of nets [36–40]. Study responses and our observations indicate that village bedframes often have jagged edges that increase the risk of rips in ITNs. When ITNs get dirty, people will wash the nets to improve their appearance in the home. However, some of the washing and drying techniques are abrasive and it is likely that many nets are overwashed. Previous studies verify that misunderstandings about ITN washing can damage the nets [41–45]. Future alterations of ITNs or community interventions that decrease over-washing will increase durability of the nets.

In addition, participants regularly discussed other potential solutions, such as village educators. One example of the success in community engagement to fight malaria is PECADOM PLUS (Prise en charge à domicile), which is a Senegalese Ministry of Health developed program that Peace Corps volunteers help to extend into the rural communities [46]. Community participation like this has been found to reduce disease burden [47–51]. Maintaining a high level of community involvement is also very important to sustain high levels of ITN use where malaria rates are decreasing [52–54].

It is important to note that the failure to achieve optimal protection with nets in these villages is not related to a lack of acceptability of ITNs or a lack of awareness about malaria. This study's responses were saturated with positive comments about nets. Participants demonstrated knowledge about the deadly nature of malaria, the vulnerability of children, the economic benefits of avoiding illness, and even the life cycle of *Plasmodium*.

## Conclusions

In a rural Senegalese population with a high malaria burden, our survey indicated a need for 0.86 insecticide treated nets per person. This is 54% higher than the current WHO recommendation of 0.56 ITNs per person.

Our findings suggest that there are not enough nets because routine village conditions lead to considerable net damage, and because the heat leads people to sleep outdoors, where they likely do not have mosquito nets. With global warming, we suspect this population will spend even more time sleeping outside, aggravating the current insufficiency. To help accelerate the decline of malaria mortality, further research should investigate optimal interventions to address this challenge, including nets designed for outside use and for higher durability.

## Supporting information

**S1 Fig. Children at shade structure.** An outdoor shade structure, used during the day for food preparation. In the evening, this structure was a hub for making tea and conversation. For the first half of the night, this structure was regularly used for sleeping. Reprinted from the author's personal collection, with permission from Andrew Sherman, without copyright restrictions.
(ZIP)

**S2 Fig. Outdoor sleeping space.** A rare example of outdoor structures with mosquito nets. These elevated sleeping platforms were made with long wood sticks, which typically had sharp ends which catch and tear ITNs. Reprinted from the author's personal collection, with permission from Andrew Sherman, without copyright restrictions.
(ZIP)

**S3 Fig. Boy by outdoor sleeping space.** Another example of a typical outdoor structure made with bamboo-like grass. Here, the sharp ends of this material are evident. Reprinted from the author's personal collection, with permission from Andrew Sherman, without copyright restrictions.
(ZIP)

**S4 Fig. Outdoor structure used for food preparation.** This outdoor structure (left of the hut) was commonly used for drying grain on its top level. Its top level was made from a piece of local wooden fencing, also with sharp edges. Reprinted from the author's personal collection, with permission from Andrew Sherman, without copyright restrictions.
(ZIP)

**S1 File. Survey tool (French).** After piloting, the French version of this study's survey was created.
(PDF)

**S2 File. Survey tool (English).** The English version of the survey, back translated from French.
(PDF)

**S3 File. Survey tool (Pulaar/Fulani).** The local language translation of the study survey.
(PDF)

**S4 File. Video outdoor sleeping.** Discussion of outdoor sleeping space usage. Reprinted from the author's personal collection, with permission from Andrew Sherman, without copyright restrictions.
(ZIP)

**S1 Checklist. Questionnaire on inclusivity in global research.**
(DOCX)

## Acknowledgments

This work could not be done without the lifelong efforts of Bandafassi health post nurse, Mactar Mansaly. Thank you to all the assisting professionals at the University of Rochester Department of Pediatrics. Thank you to Anne Longtine MD and Carrie Vargo MD. Thank you to the Bandafassi area village health workers, the people of Thioke Thian, the local Peace Corps Volunteers, the members of the Programme National de Lutte contre le Paludisme, and the Senegal Ministry of Health.

## Author contributions

**Conceptualization:** Andrew C.L. Sherman, C. Andrew Aligne, Jesse D. Matthews.

**Data curation:** Andrew C.L. Sherman.

**Formal analysis:** Andrew C.L. Sherman.

**Investigation:** Andrew C.L. Sherman, C. Andrew Aligne, Jesse D. Matthews.

**Methodology:** Andrew C.L. Sherman, C. Andrew Aligne, Jesse D. Matthews.

**Project administration:** Andrew C.L. Sherman, C. Andrew Aligne.

**Resources:** Andrew C.L. Sherman.

**Supervision:** Andrew C.L. Sherman, C. Andrew Aligne.

**Validation:** Andrew C.L. Sherman, C. Andrew Aligne.

**Visualization:** Andrew C.L. Sherman, Jesse D. Matthews.

**Writing – original draft:** Andrew C.L. Sherman.

**Writing – review & editing:** C. Andrew Aligne, Jesse D. Matthews.

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
