## [Decision Letter · Decision Letter 0]

29 Jan 2025

Dear Dr. Sherman,

Thank you for submitting your manuscript to PLOS ONE. After careful consideration, we feel that it has merit but does not fully meet PLOS ONE’s publication criteria as it currently stands. Therefore, we invite you to submit a revised version of the manuscript that addresses the points raised during the review process.

We look forward to receiving your revised manuscript.

Kind regards,

Rajib Chowdhury, M.Sc.; MPH

Academic Editor

PLOS ONE

Journal Requirements:

2. In the ethics statement in the Methods, you have specified that verbal consent was obtained. Please provide additional details regarding how this consent was documented and witnessed, and state whether this was approved by the IRB

3. Please include a complete copy of PLOS’ questionnaire on inclusivity in global research in your revised manuscript. Our policy for research in this area aims to improve transparency in the reporting of research performed outside of researchers’ own country or community. The policy applies to researchers who have travelled to a different country to conduct research, research with Indigenous populations or their lands, and research on cultural artefacts. The questionnaire can also be requested at the journal’s discretion for any other submissions, even if these conditions are not met.  Please find more information on the policy and a link to download a blank copy of the questionnaire here: https://journals.plos.org/plosone/s/best-practices-in-research-reporting. Please upload a completed version of your questionnaire as Supporting Information when you resubmit your manuscript.

5. Please provide a complete Data Availability Statement in the submission form, ensuring you include all necessary access information or a reason for why you are unable to make your data freely accessible. If your research concerns only data provided within your submission, please write "All data are in the manuscript and/or supporting information files" as your Data Availability Statement.

6. We note that Figure 2 in your submission contain [map/satellite] images which may be copyrighted. All PLOS content is published under the Creative Commons Attribution License (CC BY 4.0), which means that the manuscript, images, and Supporting Information files will be freely available online, and any third party is permitted to access, download, copy, distribute, and use these materials in any way, even commercially, with proper attribution. For these reasons, we cannot publish previously copyrighted maps or satellite images created using proprietary data, such as Google software (Google Maps, Street View, and Earth). For more information, see our copyright guidelines: http://journals.plos.org/plosone/s/licenses-and-copyright.

Reviewers' comments:

Reviewer's Responses to Questions

**Comments to the Author**

1. Is the manuscript technically sound, and do the data support the conclusions?

Reviewer #1: Yes

Reviewer #2: Partly

Reviewer #3: Partly

Reviewer #4: Partly

2. Has the statistical analysis been performed appropriately and rigorously?

Reviewer #1: Yes

Reviewer #2: No

Reviewer #3: N/A

Reviewer #4: N/A

3. Have the authors made all data underlying the findings in their manuscript fully available?

Reviewer #1: Yes

Reviewer #2: No

Reviewer #3: Yes

Reviewer #4: No

4. Is the manuscript presented in an intelligible fashion and written in standard English?

Reviewer #1: Yes

Reviewer #2: No

Reviewer #3: Yes

Reviewer #4: Yes

Reviewer #1: The authors set out to assess the effect of climate change on malaria prevention using Insecticide treated bed-nets (ITNs) in a high malaria transmission rural area of Senegal. They used a structured questionnaire designed according to the Peace Corps model to conduct a survey in the designated area. They also employed focused interviews to complement the surveys. Although ethical clearance for the survey was obtained at the University of Rochester ,USA, the authors did not indicate whether they also obtained ethical clearance from health authorities in Senegal. If they did, they should indicated this in the revised manuscript. If they did not they should explain why this was not done.

The main finding of the study is that the use of ITNs has reduced because of increased ambient temperature that makes their use uncomfortable for those surveyed. The authors confirmed that the surveyed population were well informed about malaria and its prevention by the use of ITNs. The rise of temperature mentioned by the authors is not supported by meteorological data from the area studied. Usually such records are kept in francophone African countries. The authors should include supporting data to the claim of climate change and /or variability in the study area.

I found the following stylistic errors which the authors should correct before the manuscript is published:

1. line 166: Write "All the 164 selected households..."instead of "164 of the164 households..."

2.line 171: Write, "health care facility/post/centre)'instead of healthcare access...'

3.line 182: Write "What do you think...?" instead of "How do you think..."

4 The Discussion should be revised to interpret the results, not to restate them as it currently does. The authors should discuss alternative and/or supplementary methods of malaria prevention such as indoor spraying with insecticides and the use of mosquito traps.

5.The Conclusion is long and repetitive. Please reduce it to 2-3 sentences highlighting the take home message of the study.

Notwithstanding the minor concerns stated above the paper is well- researched, well written and provides useful information on malaria prevention in a high-transmission rural area of sub-Saharan Africa. The manuscript should be accepted for publication in PLoS ONE after addressing the issues raised above.

Reviewer #2: Please read carefully, I have attached the word file follow the suggestions and recommendations.

Review report for PLOS ONE #PONE-D-24-47493 Research Article

Journal: PLOS ONE

Dated: 3rd January, 2025

Title: Malaria prevention in the age of climate change: A community survey in rural Senegal

Short Title: Malaria prevention survey in rural Senegal

Title.

1. Required minor change if needed. In rural areas is more pronounced or Senegal countryside??

Introduction

1. Introduction is very short.

2. Write down about mosquito specie names, and their biology, ecology and dispersion across globe.

3. Which species cause malaria and other associated diseases.

4. Lack of scientific writing in the whole manuscript. Need improvement

5. Aims and objectives are ok.

Materials and Methods

1. Line 94 Study population, should be on the start of M&M

2. Lack of required for study

3. Lack of primary & secondary outcomes.

4. Hypothesis.

Methods

1. Lack of research question improve more!

2. There are no new technique/methods

3. Outcomes is confused

4. Write down the “Statistical analysis” in the M&M section.

Results

1. Lack of demographic presentation.

2. Lack of data presentation

3. Tables & figures need to improve

4. Validity results internal need improve

5. Validity results external require improve

Discussion

1. Write down the discrimination statistical and significant studies.

2. Write down the clear findings

3. Need to write comparison with previous research.

4. Discuss strengths of research

5. Write down limitation of research

Conclusion

1. Need reference application of pesticides.

2. Need scientific writing.

3. Global warming reference.

4. Conclusion based on the result so in the manuscript there were no discuss about CO2.

My suggestion:

1. Major revision.

2. Check the whole manuscript text wording fonts or formatting.

3. Overall quality of manuscript is good.

4. Need clearer in English writing, grammatical mistakes and preposition etc.

5. Introduction need more improvement

6. Materials and methods needs in the survey questioner would be more interesting in table form

7. Results section the writing of the outcomes is very difficult to understand either would be in table form. There are too may quotation, apostrophe/ punctuation (‘’) marks in the results and discussion section.

8. Discussion,

9. Conclusions need more clear

10. References.

Reviewer #3: In my humble opinion, the prepared manuscript is decent and highlights one of the important concerns in public health, especially for Africa. However, the manuscript falls short on many aspects. These are as below:

1. The authors have documented that there had been reductions in malaria cases and deaths between the periods of 2010-2015; the estimate have now increased in 2025. The conductance of this research was in 2012, a period where one would expect malaria cases to decrease. Important questions that arise what was the prevalence of malaria then and what is now? How many died then and how many now? What were the factors contributing to the change in malaria cases then, and what now? Such comparisons appear to be missing.

2. To my understanding, Africa already has a hot climate. Global warming does affect change in temperatures (increasing by 1.5-2 degrees Celsius, every year). An important question arises is sleeping outdoors a norm in Africa, or has the habit changed over time? If the habit has evolved overtime, are the average temperatures remarkably different to what were prior to 2012 and after?

3. The main focus of the authors appears more on the insecticide treated mosquito nets (ITNs). In 2012, one would expect the ITNs being one of the important measures contributing towards lowering the cases of malaria in the region. Other measures may also be as important to contribute to reductions in malaria cases (especially in 2012), such as insecticide spraying, using mosquito repellants, and covering of exposed parts of the body. Are there any views on how the interviewed village folk protected themselves using other measures? Are these still in practice?

4. A question also to ponder is that the use of ITNs is “relatively easy” within households as structures are already in place to hang these nets on the bed or other sleeping places. Are there any views on “erecting” structures so that ITNs could be hanged over sleeping places outdoors?

5. The authors have highlighted an important aspect of ITNs; its quality to be durable and long-lasting. From the views of the interviewed population, the ITNs do not appear to be durable. To my understanding WHO has issued guidelines (from 2011) to monitor the durability of ITNs; to withstand about 20 washes, lifespan to be from 1-3 years. Are there any views as to how many times the ITNs had been washed or how long a net had been used by the interviewed population? Is such a practice still being seen in 2023?

6. Conclusions drawn by the authors indicate that the population has a pretty sufficient number of ITNs than what was recommended by WHO. However, this estimate appears more valid for 2012 (and likely contributing to the reductions in malaria cases and deaths). The authors do indicate follow-ups in 2023 also. It will be of much public health importance to compare the perceptions and views of the population of Senegal in 2012 and 2023, which, in turn, will also assist in identifying the factors responsible for malaria resurgence. In view of the ITNs, durability could be explored. One plausible factor for resurgence could be resistance of mosquitoes to the insecticides and repellants. Climate change might also have an impact, however, this need to be explored in greater detail in view of climatic factors such as fluctuations and trends in temperature and precipitation, influencing the rates of malaria and its vector.

Minor edits:

1. Caution needs to be exercised while referencing. For example, in para 1 of the introduction, estimates need to be properly referenced.

Reviewer #4: - Data collection is 12 years back. How is the results of the study useful in current situation of Malaria in Senegal?

- Sentence in lines 63 to 65 need to be rephrased.

- The rationale of the study is not clear

- Paragraph start line 122 can be merged with the paragraph start in line 126 as information seems repeated.

- Line 172-173 not clear, what does the average population stand for?

- Discussion part can be more precise

- Conclusion is too long and reiteration of the information already mentioned in Discussion part

- Figure quality poor

**Do you want your identity to be public for this peer review?** For information about this choice, including consent withdrawal, please see our Privacy Policy

Reviewer #1: No

Reviewer #2: No

Reviewer #3: No

Reviewer #4: No

---

## [Author Response · Author response to Decision Letter 1]

30 Mar 2025

Please refer to the Response to Reviewers attached document.

Again, thank you for all your time and work on this.

---

## [Decision Letter · Decision Letter 1]

23 Apr 2025

Dear Dr. Sherman,

Thank you for submitting your manuscript to PLOS ONE. After careful consideration, we feel that it has merit but does not fully meet PLOS ONE’s publication criteria as it currently stands. Therefore, we invite you to submit a revised version of the manuscript that addresses the points raised during the review process.

We look forward to receiving your revised manuscript.

Kind regards,

Rajib Chowdhury, M.Sc.; MPH

Academic Editor

PLOS ONE

**Journal Requirements:**

Reviewers' comments:

Reviewer's Responses to Questions

**Comments to the Author**

Reviewer #1: All comments have been addressed

Reviewer #4: All comments have been addressed

2. Is the manuscript technically sound, and do the data support the conclusions?

Reviewer #1: Yes

Reviewer #4: Partly

3. Has the statistical analysis been performed appropriately and rigorously?

Reviewer #1: Yes

Reviewer #4: N/A

4. Have the authors made all data underlying the findings in their manuscript fully available?

Reviewer #1: Yes

Reviewer #4: Yes

5. Is the manuscript presented in an intelligible fashion and written in standard English?

Reviewer #1: Yes

Reviewer #4: (No Response)

**Reviewer #1: ** I have looked through the authors' responses to. my concerns and I am fully satisfied with them.Itherefore recommend the paper be accepted.

**Reviewer #4:**  I appreciate the author to adequately addressing reviewers comments. I have few suggestions that may help improvise the manuscript.

1. Please check and rephrase the sentences that started with numbers, which is not in a general practice. For example in line 42, 45 and 255.

2. Paragraph starts from line 118 to 121, the hypothesis may not be required here.

3. In the methodology section, study population description can be concise.

4. Lines 139 to 141 can be rephrased or omitted.

5. Lines 196 to 203 can be concise.

6. In conclusion, be precise about the take home message.

**Do you want your identity to be public for this peer review?** For information about this choice, including consent withdrawal, please see our Privacy Policy

Reviewer #1: No

Reviewer #4: **Yes: ** Lalita Roy

---

## [Author Response · Author response to Decision Letter 2]

15 May 2025

Please see the Response to Reviewers file for our responses to any recent questions or recommendations. Thank you for your time and work on this process.

---

## [Decision Letter · Decision Letter 2]

12 June 2025

Malaria prevention in the age of climate change: A community survey in rural Senegal

PONE-D-24-47493R2

Dear Dr. Sherman,

We’re pleased to inform you that your manuscript has been judged scientifically suitable for publication and will be formally accepted for publication once it meets all outstanding technical requirements.

Kind regards,

Rajib Chowdhury, M.Sc.; MPH

Academic Editor

PLOS ONE

Additional Editor Comments (optional):

Reviewers' comments:

Reviewer's Responses to Questions

**Comments to the Author**

Reviewer #1: All comments have been addressed

Reviewer #2: All comments have been addressed

Reviewer #4: All comments have been addressed

2. Is the manuscript technically sound, and do the data support the conclusions?

Reviewer #1: Yes

Reviewer #2: Partly

Reviewer #4: Yes

3. Has the statistical analysis been performed appropriately and rigorously?

Reviewer #1: Yes

Reviewer #2: I Don't Know

Reviewer #4: N/A

4. Have the authors made all data underlying the findings in their manuscript fully available?

Reviewer #1: Yes

Reviewer #2: Yes

Reviewer #4: Yes

5. Is the manuscript presented in an intelligible fashion and written in standard English?

Reviewer #1: Yes

Reviewer #2: Yes

Reviewer #4: Yes

Reviewer #1: This article provides valuable insights to the malaria control strategy using insecticide treated bed nets in a rural setting of Senegal. The revised article has satisfactorily addressed my concerns expressed previously, including about ethical clearance for the study. Therefore, I recommend that the manuscript be accepted for publication in PLoS ONE.

Reviewer #2: (No Response)

Reviewer #4: I would appreciate all the responses made, though not fully satisfied. I hope the finding would guide to fight against Malaria in hard to reach settings in context of climate change.

**Do you want your identity to be public for this peer review?** For information about this choice, including consent withdrawal, please see our Privacy Policy

Reviewer #1: No

Reviewer #2: No

Reviewer #4: No

---

## [Editor Report · Acceptance letter]

PONE-D-24-47493R2

PLOS ONE

Dear Dr. Sherman,

I'm pleased to inform you that your manuscript has been deemed suitable for publication in PLOS ONE. Congratulations! Your manuscript is now being handed over to our production team.

Kind regards,

on behalf of

Dr. Rajib Chowdhury

Academic Editor

PLOS ONE